# Denoising Autoencoder-Based Feature Extraction to Robust SSVEP-Based BCIs

**DOI:** 10.3390/s21155019

**Published:** 2021-07-23

**Authors:** Yeou-Jiunn Chen, Pei-Chung Chen, Shih-Chung Chen, Chung-Min Wu

**Affiliations:** 1Department of Electrical Engineering, Southern Taiwan University of Science and Technology, Tainan 71005, Taiwan; chenyj@stust.edu.tw (Y.-J.C.); chung@stust.edu.tw (S.-C.C.); 2Department of Mechanical Engineering, Southern Taiwan University of Science and Technology, Tainan 71005, Taiwan; chenpc@stust.edu.tw; 3Department of Intelligent Robotics Engineering, Kun-Shan University, Tainan 710303, Taiwan

**Keywords:** denoising autoencoder, steady state visually evoked potential, brain computer interface, noise suppression, deep neural network

## Abstract

For subjects with amyotrophic lateral sclerosis (ALS), the verbal and nonverbal communication is greatly impaired. Steady state visually evoked potential (SSVEP)-based brain computer interfaces (BCIs) is one of successful alternative augmentative communications to help subjects with ALS communicate with others or devices. For practical applications, the performance of SSVEP-based BCIs is severely reduced by the effects of noises. Therefore, developing robust SSVEP-based BCIs is very important to help subjects communicate with others or devices. In this study, a noise suppression-based feature extraction and deep neural network are proposed to develop a robust SSVEP-based BCI. To suppress the effects of noises, a denoising autoencoder is proposed to extract the denoising features. To obtain an acceptable recognition result for practical applications, the deep neural network is used to find the decision results of SSVEP-based BCIs. The experimental results showed that the proposed approaches can effectively suppress the effects of noises and the performance of SSVEP-based BCIs can be greatly improved. Besides, the deep neural network outperforms other approaches. Therefore, the proposed robust SSVEP-based BCI is very useful for practical applications.

## 1. Introduction

Amyotrophic lateral sclerosis (ALS), which causes an interruption of the output of the central nervous system to the muscles, would degrade the communication ability [1,2]. Thus, a subject with ALS will no longer be able to communicate with others or devices without assistance. Since ALS does not affect the sensory nerves and the autonomic nervous of a subject with ALS, the steady state visually evoked potential (SSVEP)-based brain computer interfaces (BCIs), which are independent of muscle control, are very suitable for implementing an alternative augmentative communication (AAC). However, the noises, which are always appeared and acquired for the practical applications, would severely degrade the performance of SSVEP-based BCIs. Thus, developing a robust SSVEP-based BCI is very important for practical applications.

Subjects with ALS can rely on AAC to facilitate communication [3,4,5,6,7]. Hornero et al. developed a communication board to help subjects with speech disabilities [3]. Using this AAC device, the subjects can touch the command sheets on the communication board to pronounce a specified speech. To help subjects with severe disabilities, Jafari et al. proposed a tongue drive system, which uses voluntary tongue movements as the input interface, to help people with accessing their environment [4]. Anila and Radhika used the Morse code, which is detected from the lip contour, as a human communication interface [5]. Thus, the patients can easily communicate with others when they are familiar with the Morse code. Garcia et al. proposed a wearable AAC device to help subjects identify the pre-defined words, which are adopted to present the specified needs, by using the information of discrete breathing patterns [6]. Radici et al. designed an AAC app, which uses a speech symbol technology, to express complex communication needs [7]. However, operating AAC systems is dependent on muscle control, which is very difficult for subjects with ALS. Therefore, developing an interface, which does not use muscle control, can effectively ease the communication of subjects.

BCIs had been widely developed to allow subjects to control devices or communicate with others by modulating their brain signals [8,9,10,11,12,13,14,15,16,17,18]. Generally, the BCIs use near-infrared spectroscopy, functional magnetic resonance imaging, magnetoencephalography, or electroencephalogram (EEG) to monitor a user’s brain activities [8,9,10]. Using the EEG as the recording methods has a relatively lower cost [11,12], thus the EEG had been successfully used in BCIs. For electrical BCIs, SSVEP, motor imagery, and P300 potentials had been widely used to represent the results of brain activity. For SSVEP-based BCIs, a visual stimulus with a specific frequency is applied to evoke the specified electrical activities, and then, the EEG signals can be recorded. The frequency of the elicited SSVEP signal should be the same with the multiples of the frequency of the visual stimulus. In the last decades, many researches had shown that the SSVEP-based BCIs can achieve an excellent signal-to-noise ratio [13,14]. Therefore, the signal stability of SSVEP-based BCIs is better than other approaches [15]. Thus, the complexity of the signal process can be effectively reduced and it is suitable to develop practical applications. However, for the practical applications, the EEG signals always contain noises and the performances of SSVEP-based BCIs are severely degraded [16,17]. Thus, developing robust SSVEP-based BCIs allows to increase the performance and the values of SSVEP-based BCIs.

Recently, many noise suppression approaches have been proposed to improve the performance of applications, especially for speech or image applications [19,20]. Moreover, deep learning approaches always outperform other traditional approaches. For denoising autoencoder-based neural network, the inputs are perturbed by artificial noise and then, the neural network is trained to remove the noisy components for constructing clean outputs. Many applications showed that using denoising autoencoder-based neural networks can achieve acceptable results of noise reduction. Therefore, the denoising autoencoder-based neural network would be very useful in developing a robust SSVEP-based BCI.

In this study, a robust SSVEP-based BCI is proposed to help subjects communicate with others or devices. To effectively elicit the SSVEP signal, the visual stimuli with specific frequencies are displayed on an LCD monitor. To precisely represent the characteristics of SSVEP signals, the denoising autoencoder-based neural network is proposed to extract the denoising features. To correctly find the results, deep neural networks (DNN) are adopted as the decision models for finding the commands of a subject.

The rest of this paper is organized as follows. The proposed robust SSVEP-based BCI is described in Section 2. Section 3 then presents a series of experiments conducted to evaluate the performance of our approach. Conclusions and recommendations for future research are finally drawn in Section 4.

## 2. Robust SSVEP-Based BCIs

The flowchart of proposed robust SSVEP-based BCI using denoising autoencoder-based neural networks and DNN is shown in Figure 1. First, the visual stimuli with different flicking frequencies are displayed on the LCD monitor and then, a subject uses the visual stimulus to elicit the corresponding EEG signals. Second, the elicited EEG signals are acquired and the denoising autoencoder-based neural network is designed and used to extract the corresponding robust features. Finally, a DNN is adopted to identify the decisions, which are used to represent the designed commands or messages. These procedures are detailed in the following subsections.

### 2.1. Visual Stimulation and SSVEP Signal Acquisition

In this study, five blinking boxes were designed as the visual stimuli and used to elicit the SSVEP signal of a subject. Therefore, only five commands were assigned to these five blinking boxes and selected by a subject. The five blinking boxes were displayed on the 20″ LCD monitor and placed as pentagons for effectively reducing the interference between each visual stimulus. Since the refresh rate is 60 Hz, the blinking frequencies for the five blinking boxes were 6.00 Hz, 6.67 Hz, 7.50 Hz, 8.57 Hz, and 10.00 Hz [13].

The subjects were asked to sit in front of the LCD monitor and the distance measured from the subject’s nasion to the monitor was 55 cm. A NuAmps EEG amplifier, which was supplied by the Neuroscan Company, was used to acquire the elicited SSVEP signals by using Neuroscan Quickcap electrode cap with 40 channels. The EEG signals were then acquired from the Oz channel, which was connected to the visual cortex of the brain. The reference and ground electrodes were placed at A1 and A2.

### 2.2. Robust Feature Extraction

The flowchart of feature extraction by using denoising autoencoder-based neural networks is presented in Figure 2. An acquired EEG signal *x*(*t*) is the sum of an ideal SSVEP signal, *s*(*t*), and a noise signal *n*(*t*) and it can be written as
(1)x(t)=s(t)+n(t).

However, the ideal SSVEP signal cannot be obtained. In this study, *s_i_*(*t*) of the *i*-th visual stimulus was assumed as a sine wave and it can be defined as
(2)$$s_{i}\left(t\right)=A\sin\left(2\pi f_{i}t+\varphi_{i}\right),$$
where *A*, *f_i_*, and *φ_i_* are amplitude, ordinary frequency and phase, respectively. In the training stage, the cross correlation was adopted to estimate *φ_i_* from *x*(*t*) and a sine wave.

The denoising autoencoder-based neural network would estimate a denoising SSVEP signal $x^{\prime }\left(t\right)$ such that $x^{\prime }\left(t\right)$ is very similar to *s*(*t*) and *n*(*t*) in Equation (1) can be effectively suppressed. First, the denoising autoencoder-based neural network would use a deterministic mapping function *M_θ_* = {*W*, *b*} to map *x*(*t*) to a hidden representation *y*(*t*). *W* and *b* are the weight matrix and the bias vector. In this study, the *y*(*t*) was adopted as the robust feature and it can be written as
(3)$$y\left(t\right)=M_{\theta }\left(x\left(t\right)\right)=Wx(t)+b,$$

Second, denoising autoencoder neural network attempts to reconstruct $x^{\prime }\left(t\right)$ via a reconstruction mapping function ${M^{\prime }}_{\theta^{\prime }}=\left\{M^{\prime },b^{\prime }\right\}$. Thus, $x^{\prime }\left(t\right)$ can be obtained and written as
(4)$$x^{\prime }\left(t\right)={M^{\prime }}_{\theta^{\prime }}\left(y\left(t\right)\right)=W^{\prime }y(t)+b^{\prime },$$

Finally, the traditional squared error is adopted as the loss function *L* in this study. Therefore, the parameters *θ* and *θ*′ can be estimated by minimizing reconstruction errors and written as
(5)$$\begin{array}{cc} \hat{\theta },{\hat{\theta }}^{\prime } & =\underset{\theta ,\theta^{\prime }}{\arg\min}\frac{1}{K}\sum_{i=1}^{K}L\left({x^{\prime }}_{i}\left(t\right),s_{i}\left(t\right)\right)\\ & =\underset{\theta ,\theta^{\prime }}{\arg\min}\frac{1}{K}\sum_{i=1}^{K}L\left({M^{\prime }}_{\theta^{\prime }}\left(M_{\theta }\left(x\left(t\right)\right)\right),s_{i}\left(t\right)\right) \end{array}$$
where *K* is the number of training samples.

### 2.3. Deep Neural Network-Based Response Recognition

The DNN, which is a standard feed-forward fully connected neural network, is adopted as the recognition model and is illustrated in Figure 3. The input of DNN is the robust features extracted from the denoising autoencoder-based neural network. For each hidden and output layer, the weighted sum *z* of the inputs, which are the outputs of previous neurons, is computed. Then, the activation function used in this study is a parametric rectified linear unit *f*(*z*), which is defined as
(6)$$f\left(z\right)=\{\begin{array}{cc} 0, & \mathrm{if}\;z\leq 0\\ z, & \mathrm{if}\;z>0 \end{array},$$

To train the DNNs, the back-propagation algorithm, which is the most widely used approach, is applied to update the parameters of DNNs. In the back-propagation algorithm, the gradient of prediction loss is computed in one layer at a time, then it iterates backward from the output layer through the entire network. The training process of DNN is provided as follows.

Step 1. Randomly select the input data *y_n_*, which is obtained from the denoising autoencoder-based neural network.Step 2. Generate the corresponding target data of output layer ${\hat{o}}_{n}$.Step 3. For *y_n_*, the corresponding output can be obtained from the output layer and the Euclidean loss function is selected and defined as
(7)$$E=\frac{1}{2N}\sum_{n=1}^{N}\left\|{\hat{o}}_{n}-o_{n}\right\|,$$Step 4. According to the loss and the back-propagation algorithm, the parameters of DNN are updated as
(8)$$w\left(i+1\right)=w\left(i\right)-\eta \frac{\partial E}{\partial w\left(i\right)},$$
where *w*(*i*) is the weight at *i*-th iteration and *η* is the learning rate.Step 5. Repeat step 3 to step 4 until the loss is minimized.

## 3. Experimental Results and Discussions

To evaluate the proposed robust SSVEP-based BCI, a visual stimulation procedure with 5 sets of stimulation sequences is designed. Each set of stimulation sequences consists of 3 stimulation frequencies, which were randomly selected from the given 5 frequencies. Each set of stimulation sequences follows the procedure: each set begins with a 5 s countdown delay, then it is followed by a series of 10 s of visual stimulation and 10 s rest. Afterward, one minute of compulsory rest time is provided for the subject after every set of stimulation sequences. The acquired EEG signals are then blocked into 10 non-overlapping frames. The duration for a segment is one second, and the sampling rate is 100 Hz.

In this study, 15 healthy subjects (11 males and 4 females) aged between 21 and 23 years were asked to participate in the experiments and they signed the agreements to attend the test of the project. The subjects did not have previous experience using SSVEP-based BCIs and were asked to collect data in three days. Leave-one-out cross validation was used to objectively evaluate the proposed robust SSVEP-based BCI. Therefore, a subject was left as the testing data set and then others were treated as the training data set. In the following subsections, the detailed results of the proposed robust SSVEP-based BCI are examined.

### 3.1. The Experimental Results of Noise Suppression

In this subsection, the signal-to-noise ratio (SNR) is applied to evaluate the performance of noise suppression in the time domain. Moreover, the canonical correspondence analysis (CCA), which has been widely used in SSVEP-based BCIs [21], was adopted to evaluate improvement of system recognition rate. The number of harmonics for CCA was set to be 4 in this experiment.

For designing the ideal SSVEP signals, a zero-phase sine wave was treated as the target of the denoising autoencoder-based neural network, and the network would cause enhanced SSVEP signals to be zero-phase signals (denoted as DAE_AP). In this approach, the phase of the input SSVEP signal would be adjusted and then it would increase the complexity of denoising autoencoder. To reduce the complexity of denoising autoencoder, the phase of the ideal SSVEP signal and the input SSVEP signal should be the same. Therefore, cross correlation was used to estimate the phase of the input SSVEP signal. According to the estimated phase, an ideal SSVEP signal could be generated (denoted as DAE_IP).

The number of hidden nodes for denoising autoencoder was examined and the results measured by using SNR and recognition rate are shown in Table 1 and Table 2. According to the results, when the number of hidden nodes is 25, DAE_AP and DAE_IP can obtain an acceptable performance. When the number of hidden nodes is increased, the performances measured by using SNR and recognition rate are slightly improved. Therefore, when the number of hidden nodes is 25, the proposed approach can balance the computational complexity and accuracy. Moreover, the recognition rates for stimuli with different flicking frequencies are shown in Table 3. It is clear that the recognition rates of DAE_AP and DAE_AP are greatly improved, compared with that of original SSVEP signals. Thus, the effects of noises can be effectively reduced by using denoising autoencoder-based neural networks. Besides, the worst recognition rate for DAE_IP is 92.15% and it still can obtain an acceptable performance for practical applications.

### 3.2. The Experimental Results of DNN

In this subsection, the CNN is selected as the baseline system for comparison. The DNN-based classifier, whose features are extracted by using DAE_AP and DAE_IP, is examined. The experimental results are shown in Table 4. The results showed that the performances of DNN are higher than that of CNN and the error reduction rate is 54.81%. Therefore, the DAE_AP and DAE_IP can effectively remove the effects of noises and they are very useful in developing robust SSVEP-BCIs. Besides, when the number of hidden nodes is 150, the recognition rate (95.63%) is very similar to that (95.64%) with 300 hidden nodes. DNN with 150 hidden nodes would greatly reduce the computational complexity, compared to DNN with 300 hidden nodes. Therefore, the results showed that the proposed approach is acceptable for developing an alternative augmentative communications for the practical applications.

Comparing the results in Table 4 with those in Table 1 and Table 2, it is clear that the noise suppression by using DAE_IP outperforms that by using DAE_AP. However, the performance of DAE_AP is lower than that of DAE_IP. Since the difference between DAE_AP and DAE_IP is the phase information, the effects of phase in DAE_AP and DAE_IP were examined by using bubble charts. The phase information of enhanced SSVEP signals for DAE_AP and DAE_IP are shown in Figure 4a,b, respectively. Ideally, the phase of the enhanced SSVEP signal for DAE_AP and DAE_IP should be zero and diagonal, respectively. In Figure 4, the phase information of DAE_IP is close to diagonal, but the phase information of DAE_AP is not close to zero. To detail the distortion, which is measured by using the Euclidean distance, the probability density functions of DAE_AP and DAE_IP are shown in Figure 5a,b, respectively. The experimental results showed that the distortion for DAE_AP is greater than that of DAE_IP. Thus, denoising autoencoder-based neural network cannot effectively adjust the phase of an input SSVEP signal. Therefore, DAE_IP is very useful in developing robust SSVEP-based BCIs.

### 3.3. Comparison with Other Approaches

In this subsection, the classifiers by using support vector machine (SVM) and Gaussian mixture model (GMM) are considered as baseline systems and compared with proposed approaches. Moreover, the traditional autoencoder was selected as a baseline for feature extraction. The structure of neural network for traditional autoencoder is the same as the proposed denoising autoencoder. In the training procedure, the ideal SSVEP signals are the input SSVEP signals. The experimental results are shown in Table 5. Using the features of DAE_IP, the recognition rates can be effectively improved from 90.32%, 80.64%, and 89.63% to 95.63%, 94.23%, and 94.13% for DNN, SVM, and GMM, respectively. Besides, the performances of using the features of DAE_AP are higher than those of the traditional autoencoder. Thus, the proposed DAE_AP and DAE_IP are very suitable to extract robust features for different classifiers.

Finally, the experimental results evaluated by using SNR, CCA, and different classifiers using DAE_AP and DAE_IP showed that the denoising autoencoder-based neural network can effectively reduce the effects of noises. Therefore, the proposed approaches can be adopted to extract robust features. When the recognition rates of DNN were compared with that of SVM and GMM, the experimental results showed that the proposed DNN can allow to obtain the highest results. Therefore, the architecture of DNN is very suitable in developing a classifier for SSVEP-based BCIs.

Previous research had shown that the characteristics of SSVEP signals for young subjects are different from those for elder subjects or subjects with ALS [22]. The SNR of SSVEP signals is larger than those values in elder subjects and subjects with ALS. In this study, the results show that the effects of lower SNR can be effectively reduced, thus the proposed approaches may reduce the effects of SNR for elder subjects and subjects with ALS. However, in this study, only young and healthy subjects were asked to examine the performance of the proposed approaches. This is a limitation of this study and it can be improved by extending the number and different types of users.

## 4. Conclusions

In this study, a robust SSVEP-based BCI using denoising autoencoder-based neural networks and DNN is proposed. The denoising autoencoder-based neural network can effectively extract the robust features for representing the characteristics of SSVEP signals for the practical applications. Moreover, the effects of noise components can be effectively reduced. DNN can correctly map the robust features to the decision results and the recognition rate of DNN is higher than that of SVM and GMM. The experimental results showed that the proposed approaches can effectively suppress the noises and then allow to obtain an acceptable recognition rate. Thus, the robust SSVEP-BCIs can be used for practical applications, and it can then help subjects communicate with others or devices. In the future, the elder subjects and the subjects with ALS can be asked to participate in the experiments to evaluate the value of the proposed approaches in real applications. Moreover, the different types of neural networks, such as U-Net, ResNet, MobileNet, and long short-term memory neural networks can be applied to improve the performance of classification.

## Figures and Tables

**Figure 1 sensors-21-05019-f001:**
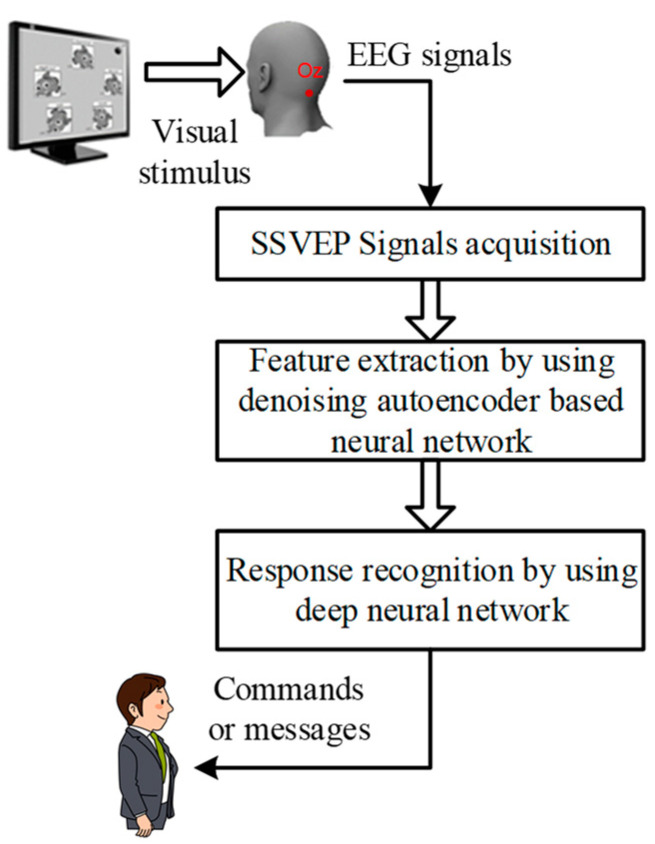
The flowchart of the proposed robust SSVEP-based BCI.

**Figure 2 sensors-21-05019-f002:**
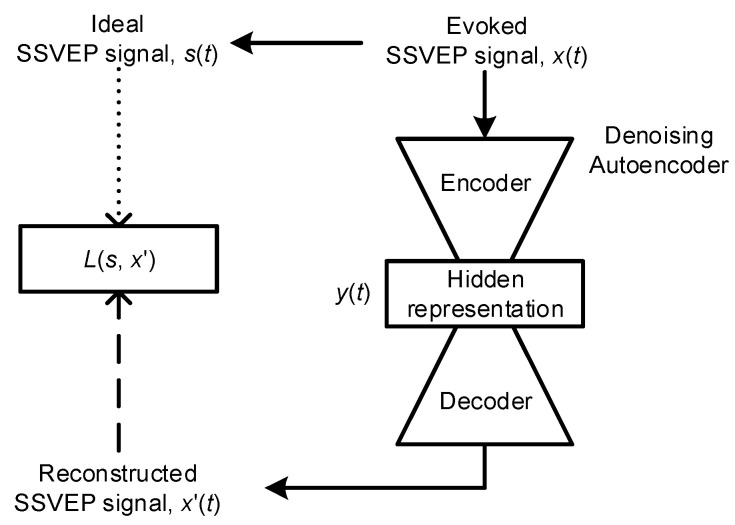
The flowchart of denoising autoencoder-based neural networks for robust feature extraction.

**Figure 3 sensors-21-05019-f003:**
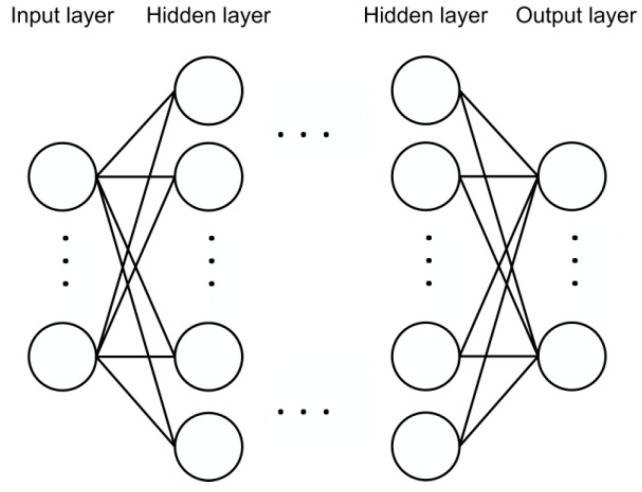
The architecture of a stacked deep neural networks.

**Figure 4 sensors-21-05019-f004:**
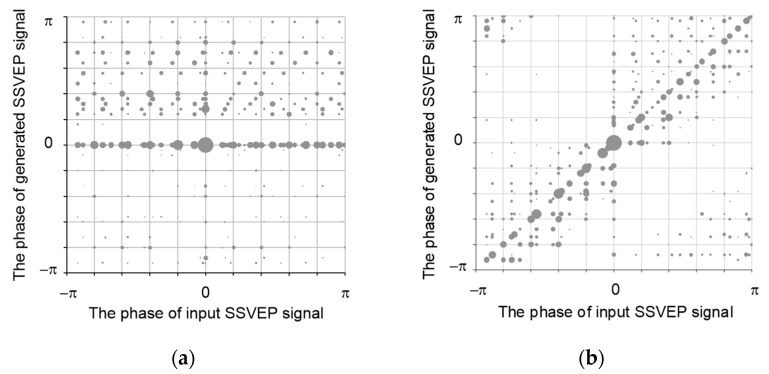
The phase information of enhanced SSVEP signal for (**a**) DAE_AP and (**b**) DAE_IP.

**Figure 5 sensors-21-05019-f005:**
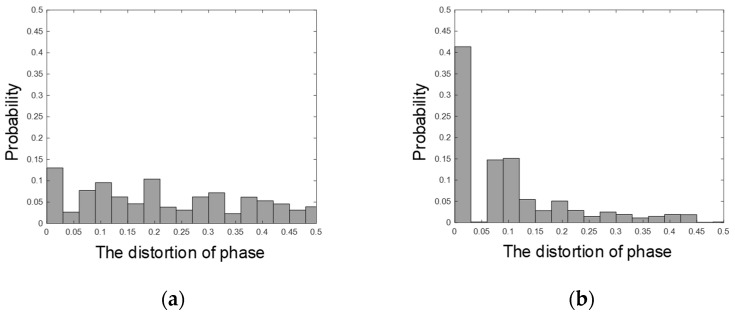
The probability density functions of phase distortions for (**a**) DAE_AP and (**b**) DAE_IP.

**Table 1 sensors-21-05019-t001:** The experimental results of DAE_AP and DAE_IP in SNR.

	The Number of Hidden Nodes		
	20	25	50
DAE_AP	1.001 dB	1.918 dB	1.974 dB
DAE_IP	1.255 dB	6.733 dB	6.884 dB

**Table 2 sensors-21-05019-t002:** The recognition rates (%) of DAE_AP and DAE_IP.

	The Number of Hidden Nodes		
	20	25	50
DAE_AP	82.51	92.04	92.84
DAE_IP	84.24	94.01	95.44

**Table 3 sensors-21-05019-t003:** The detailed recognition rates (%) for stimuli with different flicking frequencies (Hz).

	Frequency of Stimuli				
	6.00	6.67	7.50	8.57	10.00
Original SSVEP signal	93.56	91.44	93.22	89.89	86.11
DAE_AP	95.89	91.67	93.78	91.44	87.44
DAE_IP	96.82	92.82	95.71	96.15	92.15

**Table 4 sensors-21-05019-t004:** The recognition rates (%) of DNN and CNN.

The Number of Hidden Nodes	DNN		CNN
	DAE_PA	DAE_NPA	
60	77.56	78.33	72.69
90	83.11	84.88	77.11
120	89.66	91.33	86.67
150	94.31	95.63	90.33
300	94.33	95.64	90.75

**Table 5 sensors-21-05019-t005:** The detailed recognition rates (%) for stimuli with different flicking frequencies (Hz).

The Classifiers	Traditional Autoencoder	DAE_AP	DAE_IP
DNN	90.32	94.51	95.63
SVM	88.64	92.39	94.23
GMM	89.63	92.73	94.13

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
