# Peer review of "Denoising Autoencoder-Based Feature Extraction to Robust SSVEP-Based BCIs"

_sensors, 2021, doi:10.3390/s21155019_

Round 1

Reviewer 1 Report

In this manuscript a SSVEP methodology in proposed, utilizing a denoising autoencoder based neural network. The manuscript is good, well-written with appropriate structure. The objective and the results are clearly presented. However, I have a few minor and major comments to the authors:

  1. In the manuscript it is constantly written that the proposed methodology is used to “help subjects with ALS with communication” (Abstract, Lines 53, 81, 264) Are there any subjects with ALS used in this study? If no, then please remove this statement to avoid any possible misunderstanding.
  2. Lines 107-108: “The blinking frequencies for the five blinking boxes are 6.00 Hz, 6.67 107 Hz, 7.50 Hz, 8.57 Hz, and 10.00 Hz.” Why did the authors choose these frequencies from the alpha band (instead of e.g., 6, 7, 8 and 13 Hz) ? Please provide appropriate references to justify your choice.
  3. Why did the authors choose the Oz channel instead of O1 and O2?
  4. Please include a comparison table in the Discussion section with other studies addressing the same issue.
  5. Finally, my major concern is what is the added scientific value of the current paper regarding a previously published paper of one of the co-authors [1]. In [1] a convolutional denoising autoencoder based SSVEP signal enhancement method is proposed and it seems that there are also utilized data from the same 15 subjects. Therefore, I strongly express my doubts about the originality of the manuscript; also, it is bizarre why the authors are not referring to this study [1] in the proposed manuscript. Please clearly present the differences in these two methodologies.

[1] Chuang, C. C., Lee, C. C., Yeng, C. H., So, E. C., Lin, B. S., & Chen, Y. J. (2019). Convolutional denoising autoencoder based SSVEP signal enhancement to SSVEP-based BCIs. Microsystem Technologies, 1-8.

Author Response

We much thank the reviewers for the constructive comments! Based on the review results, we were able to significantly improve the manuscript. To help the editor and reviewers go over the revision, the main responses are summarized below.

Reviewer 1:

In this manuscript a SSVEP methodology in proposed, utilizing a denoising autoencoder based neural network. The manuscript is good, well-written with appropriate structure. The objective and the results are clearly presented. However, I have a few minor and major comments to the authors:

  1. In the manuscript it is constantly written that the proposed methodology is used to “help subjects with ALS with communication” (Abstract, Lines 53, 81, 264) Are there any subjects with ALS used in this study? If no, then please remove this statement to avoid any possible misunderstanding.

Responses:

Thank you very much. In this study, we did not ask subjects with ALS to participate in the experiments. Therefore, we had removed the statements “help subjects with ALS with communication.”

On page 1, line 16:

Therefore, developing robust SSVEP-based BCIs is very important to help subjects communicate with others or devices.

On page 2, line 53:

Therefore, developing an interface, which does not use muscle control, can effectively ease the communication of subjects.

On page 2, line 81:

In this study, a robust SSVEP-based BCI is proposed to help subjects communicate with others or devices.

On page 8, line 270:

Thus, the robust SSVEP-BCIs can be used for practical applications, and then it can help subjects to communicate with others or devices.

  1. Lines 107-108: “The blinking frequencies for the five blinking boxes are 6.00 Hz, 6.67 107 Hz, 7.50 Hz, 8.57 Hz, and 10.00 Hz.” Why did the authors choose these frequencies from the alpha band (instead of e.g., 6, 7, 8 and 13 Hz) ? Please provide appropriate references to justify your choice.

Responses:

Thank you. According to clinical experience, a visual range from 15-25 Hz can easily evoke an epileptic syndrome. Therefore, the selected frequencies are lower than 10 Hz in this study. For an LCD monitor, the refresh rate is typically 60 Hz; thus, the flickering boxes could flicker in a limited range of frequencies. With such a refresh rate it is possible to simulate different frequencies by displaying appropriate numbers of frames, which are images or black boxes. To justify our choice, a reference paper was cited in this manuscript and the statement was also modified.

On page 9, line 107:

Since the refresh rate is 60 Hz, the blinking frequencies for the five blinking boxes are 6.00 Hz, 6.67 Hz, 7.50 Hz, 8.57 Hz, and 10.00 Hz [13].

  1. Why did the authors choose the Oz channel instead of O1 and O2?

Responses:

Thank you. For SSVEP applications, the Oz, O1, and O2 channels can effectively measure the activities of the occipital region. However, we use only one electrode to make it easy for users to use the SSVEP applications. Therefore, the Oz channel, which can be treated as the average activities of the occipital region, is used in this study. If multi-channels are used to design the SSVEP applications, Oz, O1, and O2 channels can be selected.

  1. Please include a comparison table in the Discussion section with other studies addressing the same issue.

Responses:

Thank you for your suggestion. However, many factors would greatly affect the design of SSVEP applications. For example, the frequencies of stimulation, the number of electrodes, features, models… Therefore, using a comparison table is not able to objectively compare with previous studies. Thus, we carefully evaluate the effects of the parameters of the proposed approaches.

  1. Finally, my major concern is what is the added scientific value of the current paper regarding a previously published paper of one of the co-authors [1]. In [1] a convolutional denoising autoencoder based SSVEP signal enhancement method is proposed and it seems that there are also utilized data from the same 15 subjects. Therefore, I strongly express my doubts about the originality of the manuscript; also, it is bizarre why the authors are not referring to this study [1] in the proposed manuscript. Please clearly present the differences in these two methodologies.

[1] Chuang, C. C., Lee, C. C., Yeng, C. H., So, E. C., Lin, B. S., & Chen, Y. J. (2019). Convolutional denoising autoencoder based SSVEP signal enhancement to SSVEP-based BCIs. Microsystem Technologies, 1-8.

Responses:

Thank you for your comments. In [1], it focuses on the SSVEP signal enhancement and the major contribution is to reduce the noises in the time domain. Therefore, using a convolutional denoising autoencoder, the noise components of SSVEP signals can be effectively estimated and then reduced. Then, it can achieve more clean SSVEP signals. Thus, the performance of the convolutional denoising autoencoder is measured by using SNR and CCA. But, in this study, we focus on the performance of classification by using the robust features, which are extracted by using denoising autoencoder, and decision models, which is a stacked deep neural network. Therefore, these two approaches are different. The SSVEP signal enhancement is not used in this study, thus it is not cited.

Reviewer 2 Report

The paper presents a promising method to extract meaningful features on SSVEP-based BCIs. The methodology is scientifically sound and the proposed method convenient. The method was evaluated regarding not only the capability to improve the SNR but also the classification performance in comparison with other methods (SVM, GMM, AE). 

Moreover, although the explanation of the basics of ANN/DNN would be unnecessary in more deep learning-oriented journals, I think it could be considered appropriate for Sensors. The decision to use a leave-one-out CV is also appropriate. Also, the implementation of the DAE_IP modification was a proper decision IMO. However, some points must be improved, IMHO:

  • Although the details provided about the signal acquisition process and the feature extraction are sufficient, this is not the case with the experimental setup. For example, it is not clear the duration of the experiments, how many targets the subjects were expected to select, how it was measured, etc.
  • Moreover, more information regarding the subjects is needed. Just to mention some examples, which is the age distribution? Do they have previous experience using SVVEP-based or other BCIs? Did they repeat the experiment in several days or is was conducted just once per subject? How long it took? etc.
  • There are several typos such as the extra dot in line 51, the missing dot in lines 187 and 193, the sentence in line 190-193 seems incomplete, the word "Machines" is missing in line 236, etc.
  • 15 participants seems a very limited number to me. I completely understand that gathering this kind of data is very challenging but, in order to validate the method, more subjects or the evaluation of the method using a different dataset is needed.
  • The limitations of the method and potential improvements as future works must be included

Author Response

We much thank the reviewers for the constructive comments! Based on the review results, we were able to significantly improve the manuscript. To help the editor and reviewers go over the revision, the main responses are summarized below.

Reviewer 2:

The paper presents a promising method to extract meaningful features on SSVEP-based BCIs. The methodology is scientifically sound and the proposed method convenient. The method was evaluated regarding not only the capability to improve the SNR but also the classification performance in comparison with other methods (SVM, GMM, AE).

Moreover, although the explanation of the basics of ANN/DNN would be unnecessary in more deep learning-oriented journals, I think it could be considered appropriate for Sensors. The decision to use a leave-one-out CV is also appropriate. Also, the implementation of the DAE_IP modification was a proper decision IMO. However, some points must be improved, IMHO:

  • Although the details provided about the signal acquisition process and the feature extraction are sufficient, this is not the case with the experimental setup. For example, it is not clear the duration of the experiments, how many targets the subjects were expected to select, how it was measured, etc.

Responses:

Thank you for your valuable suggestions. The detailed experimental setup is added in the section of “3. Experimental Results and Discussions.”

On page 5, line 162:

To evaluate the proposed robust SSVEP-based BCI, a visual stimulation procedure with 5 sets of stimulation sequences is designed. Each set of stimulation sequences consists of 3 stimulation frequencies, which were randomly selected from the given 5 frequencies. Each set of stimulation sequences follows the procedure: each set begins with a 5 seconds countdown delay then follows by a series of 10 seconds of visual stimulation and 10 seconds rest. Afterward, one minute of compulsory rest time is provided for the subject after every set of stimulation sequences. The acquired EEG signals are then blocked into 10 non-overlapping frames. The duration for a segment is one second, and the sampling rate is 100 Hz.

  • Moreover, more information regarding the subjects is needed. Just to mention some examples, which is the age distribution? Do they have previous experience using SVVEP-based or other BCIs? Did they repeat the experiment in several days or is was conducted just once per subject? How long it took? etc.

Responses:

Thank you very much. The detailed information of subjects and procedures are added.

On page 5, line 171:

In this study, 15 healthy subjects (11 males and 4 females) aged between 21 and 23 were asked to participate in the experiments and then they signed the agreements to attend the test of the project. The subjects do not have previous experience using SSVEP-based BCIs and are asked to collect data in three days. Leave-one-out cross validation was used to objectively evaluate the proposed robust SSVEP-based BCI.

  • There are several typos such as the extra dot in line 51, the missing dot in lines 187 and 193, the sentence in line 190-193 seems incomplete, the word "Machines" is missing in line 236, etc.

Responses:

Thank you for your valuable comments. These typos are corrected.

On page 2, line 51:

Radici et al. design an AAC app, which uses a speech symbol technology, to express complex communication needs [7].

On page 6, line 193:

The number of hidden nodes for denoising autoencoder was examined and the results measured by using SNR and recognition rate are shown in Table 1 and Table 2.

On page 6, line 197:

Therefore, when the number of hidden nodes is 25, the proposed approach can balance the computational complexity and accuracy.

On page 8, line 243:

In this subsection, the classifiers by using support vector machine (SVM) and Gaussian mixture model (GMM) are considered as baseline systems and compared with proposed approaches.

  • 15 participants seems a very limited number to me. I completely understand that gathering this kind of data is very challenging but, in order to validate the method, more subjects or the evaluation of the method using a different dataset is needed.

Responses:

Thank you very much. It is very hard to collect this data. In this review process, we do not have sufficient time to increase the number of participants. Therefore, one-leave-out is adopted to objectively evaluate the proposed. In the future, we would like to increase the number of participants and it is discussed in future works.

On page 8, line 271:

In the future, the subjects with ALS can be asked to participate in the experiments and it can evaluate the value of the proposed approaches in real applications. Moreover, the different types of neural networks, such U-Net, ResNet, MobileNet, and long short-term memory neural networks can be applied to improve the performance of classification.

  • The limitations of the method and potential improvements as future works must be included

Responses:

Thank you very much. The limitations of the method and potential improvements are included in the conclusion.

On page 8, line 271:

In the future, the subjects with ALS can be asked to participate in the experiments and it can evaluate the value of the proposed approaches in real applications. Moreover, the different types of neural networks, such U-Net, ResNet, MobileNet, and long short-term memory neural networks can be applied to improve the performance of classification.

Round 2

Reviewer 1 Report

The authors answered all the comments and performed the appropriate changes. 

Author Response

Thank you very much.

Reviewer 2 Report

Thank you for your effort to address most of my previous comments. IMO, due to the very limited spectrum of age from the subjects (I suppose all of them are students/researchers) a further discussion about it must be included. How can this factor affect the results? Are older people expected to perform similarly? Based on my experience, young people are usually more willing to participate in this kind of studies and, also due to their experience with electronic devices, the results are usually better. This must be discussed as well. Finally, as no real patients are included in the study, this fact must be commented and discussed as an important limitation of this study.

Author Response

Responses:

Thank you very much for your valuable comments. It is very hard for us to collect the testing data from elder/ALS subjects. A previous study, which is cited in this study, had shown that the characteristics of SSVEP signals are different between normal subjects and elder/ALS subjects. The SNR of SSVEP signals for elder/ALS subjects is lower than that of normal subjects. Our proposed approach can reduce the effects of lower SNR for normal subjects. Therefore, it may also work for SSVEP signals collected from elder/ALS, and then it may increase the accuracy of SSVEP-based BCIs for elder/ALS subjects. But, we do not examine it by using real datasets, which are collected from elder/ALS subjects. It is an important limitation of this study. Thus, we had commented and discussed it in this manuscript. We also include it in the conclusion.

On page 8, line 261:

Previous research had shown that the characteristics of SSVEP signals for young subjects are different from those for elder subjects or subjects with ALS [22]. The SNR of SSVEP signals is larger than those values in elder subjects and subjects with ALS. In this study, the results show that the effects of lower SNR can be effectively reduced, thus the proposed approaches may reduce the effects of SNR for elder subjects and subjects with ALS. However, in this study, only young and healthy subjects were asked to examine the performance of the proposed approaches. This is a limitation of this study and it can be improved by extending the number and different types of users.

On page 9, line 279:

In the future, the elder subjects and the subjects with ALS can be asked to participate in the experiments and it can evaluate the value of the proposed approaches in real applications.

On page 10, line 346:

22. Hsu, H.T.; Lee, I.H.; Tsai, H.T.; Chang, H.C.; Shyu, K.K.; Hsu, C.C.; Chang, H.H.; Yeh, T.K.; Chang, C.Y.; Lee, P.L. Evaluate the Feasibility of Using Frontal SSVEP to Implement an SSVEP-Based BCI in Young, Elderly and ALS Groups, IEEE Transactions on Neural Systems and Rehabilitation Engineering, 2016, 24, 603–615.
